# MiR-106b-5p: A Master Regulator of Potential Biomarkers for Breast Cancer Aggressiveness and Prognosis

**DOI:** 10.3390/ijms222011135

**Published:** 2021-10-15

**Authors:** Paula Lucía Farré, Rocío Belén Duca, Cintia Massillo, Guillermo Nicolás Dalton, Karen Daniela Graña, Kevin Gardner, Ezequiel Lacunza, Adriana De Siervi

**Affiliations:** 1Laboratorio de Oncología Molecular y Nuevos Blancos Terapéuticos, Instituto de Biología y Medicina Experimental (IBYME), CONICET, Buenos Aires C1428ADN, Argentina; pfarre@dna.uba.ar (P.L.F.); rduca@dna.uba.ar (R.B.D.); cmassillo@dna.uba.ar (C.M.); ndalton@dna.uba.ar (G.N.D.); kgrana@dna.uba.ar (K.D.G.); 2Department of Pathology and Cell Biology, Columbia University Medical Center, 630 W. 168th Street, New York, NY 10032, USA; klg2160@cumc.columbia.edu; 3Centro de Investigaciones Inmunológicas Básicas y Aplicadas (CINIBA), Facultad de Ciencias Médicas, Universidad Nacional de La Plata, Buenos Aires B1900, Argentina; ezequiellacunza@hotmail.com

**Keywords:** breast cancer, biomarker, aggressiveness, prognosis

## Abstract

Breast cancer (BCa) is the leading cause of death by cancer in women worldwide. This disease is mainly stratified in four subtypes according to the presence of specific receptors, which is important for BCa aggressiveness, progression and prognosis. MicroRNAs (miRNAs) are small non-coding RNAs that have the capability to modulate several genes. Our aim was to identify a miRNA signature deregulated in preclinical and clinical BCa models for potential biomarker discovery that would be useful for BCa diagnosis and/or prognosis. We identified hsa-miR-21-5p and miR-106b-5p as up-regulated and hsa-miR-205-5p and miR-143-3p as down-regulated in BCa compared to normal breast or normal adjacent (NAT) tissues. We established 51 shared target genes between hsa-miR-21-5p and miR-106b-5p, which negatively correlated with the miRNA expression. Furthermore, we assessed the pathways in which these genes were involved and selected 12 that were associated with cancer and metabolism. Additionally, *GAB1*, *GNG12*, *HBP1*, *MEF2A*, *PAFAH1B1*, *PPP1R3B*, *RPS6KA3* and *SESN1* were downregulated in BCa compared to NAT. Interestingly, hsa-miR-106b-5p was up-regulated, while *GAB1*, *GNG12*, *HBP1* and *SESN1* were downregulated in aggressive subtypes. Finally, patients with high levels of hsa-miR-106b-5 and low levels of the abovementioned genes had worse relapse free survival and worse overall survival, except for *GAB1*.

## 1. Introduction

Breast cancer (BCa) is the first in incidence and the leading cause of death by cancer among women worldwide [1]. It is a heterogeneous disease, since it has a variety of histological and biological properties in terms of genetics and epigenetics, which influences its diagnosis, treatment and prognosis. Even though there are plenty of treatments for this disease, the need of new therapeutic targets is urgent, since there is no specific treatment for the most aggressive BCa types.

Currently, BCa is stratified according to its staging and to its molecular subtype; this combination will determine the patient’s therapy [2]. In terms of staging, it can be stratified from 0 to IV according to the size of the tumor (stages I–III), the presence of secondary tumors in nodes (stages II–III), their distance and the presence of metastasis in other organs (stage IV) [3]. According to its molecular profile, BCa is mostly divided according to the presence/absence of progesterone receptor (PR), estrogen receptor (ER) and/or ErbB2 receptor (Her2) [4]. 

The main subtypes are: Luminal A (PR^+^, ER^+^ and Her2^−^), Luminal B (PR^+^, ER^+^ and Her2^+^), Her2 (PR^−^, ER^−^ and Her2^+^) and triple negative (PR^−^, ER^−^ and Her2^−^), represented mostly by the basal-like sub-classification [5]. The luminal subtype is the most prevalent, with a better prognosis and several treatments available [4]. The Her2 subtype also has anti-receptor treatment, but has a worse prognosis compared to the luminal subtype. Finally, triple negative BCa has no specific treatment available thus far; therefore, it has the worse prognosis and is, therefore, the most aggressive subtype [4].

MicroRNAs (miRNAs) are small non-coding RNAs (18–22 nucleotides) capable of modulating gene-expression [6]. It was found that miRNAs play a crucial role in cancer development and progression including BCa [7,8] and can affect several hallmarks of cancer, such as cell death resistance, invasion and metastasis activation, among others [9]. In conclusion, miRNAs can play a role acting as oncomiRs or tumor suppressor miRNAs, mostly depending on their microenvironment [10].

The aim of this work was to identify a miRNA signature deregulated in breast tumors and their target genes as possible biomarkers for BCa diagnosis and/or prognosis. We compared the expression between normal breast/normal adjacent and BCa tissues of several miRNAs from mice model and human samples. Then, we focused on hsa-miR-21-5p and miR-106b-5p that shared similar profiles between human and mice and identified their target genes. Using bioinformatics approaches, we evaluated the pathways in which these genes are involved, and their relevance in normal and cancerous tissues. We also established the overall survival and relapse free survival analysis in terms of miRNA and gene expression. Finally, the correlation between the expression of miRNAs and genes and BCa aggressiveness was assessed.

## 2. Results

### 2.1. BCa Modulates miRNA Expression in 4T1 Allografts and Human Tissue Samples

In order to evaluate the expression of several miRNAs that have been reported as related to BCa development and progression [11,12,13,14,15], we developed a triple negative BCa (TNBC) mice model. Balb/c animals were randomly divided into two groups (BCa or control) and inoculated or not with 4T1 cells. Four weeks after cell inoculation, all mice were sacrificed and tissues were collected. MiRNAs from mammary gland (MG) and tumors were isolated and assessed by stem-loop RT-qPCR (Figure 1A). We found that mmu-miR-21a-5p, miR-106b-5p, miR-125b-5p and miR-221-3p were upregulated (*p*-values: <0.0001, <0.0001, 0.0011 and 0.0027, respectively) in 4T1 allografts compared with normal MG; meanwhile mmu-miR-138-5p, miR-143-3p, miR-146a-5p and miR-205-5p were downregulated (*p*-values: <0.0001, <0.0001, <0.0001 and 0.0002 respectively).

To analyze the expression profile of these miRNAs in BCa and normal adjacent tissue (NAT) from patients, available data was obtained from the UCSC Xena bioinformatic tool [16]. The information of the mature miRNAs expression obtained by RNAseq was analyzed using paired BCa-NAT from the same patient (Figure 1B). We found that hsa-125b-5p, miR-221-3p, miR-143-3p and miR-205-5p were downregulated in BCa tissue compared to NAT, while hsa-miR-21-5p and miR-106b-5p were upregulated. Since mice and human share several miRNA sequences and combining human and murine data, we found that miR-21-5p and miR-106b-5p were both up-regulated in mice and human tumors compared to MG or NAT, respectively, while miR-205-5p and miR-143-3p were both down-regulated. We next focused the analysis on these four miRNAs.

### 2.2. Hsa-miR-21-5p and miR-106b-5p Share Several Target Genes

We obtained a list of experimentally validated target (EVT) genes for hsa-miR-21-5p, miR-106b-5p, miR-205-5p and miR-143-3p, using the DIANA TARBASE v8 resource. To identify common target genes, we used Venn diagrams (Figure 2A, Appendix A). Based on this analysis, two miRNAs-target gene sets were obtained. The first one included the target genes modulated exclusively by the up-regulated miRNAs hsa-miR-21-5p and miR-106b-5p, excluding those eight genes shared with hsa-miR-143-3p and miR-205-5p. The second one included the genes modulated exclusively by the down-regulated miRNAs hsa-miR-143-3p and miR-205-5p, excluding the gene shared with hsa-106b-5p. There was only one target gene in common between hsa-miR-143-3p and miR-205-5p, while 51 target genes were overlapped between hsa-miR-21-5p and miR-106b-5p as listed in Figure 2B. We based the rest of the analysis on this 51-gene signature.

### 2.3. Hsa-miR-21-5p and miR-106b-5p Negatively Correlate with Their Target Genes

In order to determine the relevance of hsa-miR-21-5p and miR-106b-5p in human tissues, we performed principal component analysis (PCA) of the 51 genes using normalized expression data from normal mammary gland (NMG) (GTEX), NAT and BCa samples (TCGA BRCA). The first two components (Dim 1 and 2), which explained 33.4% and 9.3%, respectively, of the total variation, were plotted. We found marked differences in the overall gene expression between NMG and BCa tissues (Figure 3A, I). As shown in Figure 3A, II, the 51 genes contributed to Dim 1 in similar ways indicating that almost all the genes contributed to the discrimination between tumor and non-tumor tissues. 

Additionally, we performed single sample gene set enrichment analysis (ssGSEA) to explore whether the 51-gene signature were coordinately up- or down-regulated within the BCa or the NAT samples using TCGA BRCA dataset. We found that the 51 target genes showed a positive ssGSEA-enrichment in BCa and NAT samples (Figure 3B). In addition, a negative correlation between the ssGSEA-enrichment and the expression of hsa-miR-21-5p and miR-106b-5p was found in BCa tissue (rho = −0.1807 and −0.2876, respectively). These results suggest that hsa-miR-21-5p and miR-106b-5p might play a relevant role in the tumorigenesis, being able to distinguish normal from tumor tissue, based on their gene expression profile.

### 2.4. Hsa-miR-21-5p and miR-106b-5p Modulate Cancer and Metabolic Related Pathways

To further analyze the relevant pathways associated with the miRNA-target genes, we performed a KEGG pathway enrichment analysis (*p*-value < 0.05) for all the EVT genes of hsa-miR-21-5p, miR-106b-5p, miR-205-5p and miR-143-3p (Figure 4A, Appendix A). In particular, this analysis showed 12 common target genes, indicated with an asterisk at Figure 4A, between hsa-miR-21-5p and miR-106b-5p associated with processes related to cancer, including the MAPK signaling pathway, TGF-β signaling pathway, ERBB1 downstream pathway, mTOR signaling pathway and Wnt signaling pathway; and several metabolic processes related to the insulin signaling pathway. 

Moreover, analyzing the hsa-miR-205-5p and miR-143-3p target genes, we found several related cancer pathways, including senescence and autophagy, signaling by EGFR in cancer, microRNA regulation of DDR among others, but no genes were shared between these two miRNAs.

We next focused on the 12 genes indicated in Figure 4A: *GAB1*, *GNG12*, *HBP1*, *MEF2A*, *PAFAH1B1*, *PPP1R3B*, *RPS6KA3*, *SESN1*, *MAP3K2*, *YY1*, *FRS2* and *STAT3*. We analyzed the expression of these genes in BCa and NAT tissue (Figure 4B). We found that *GAB1*, *GNG12*, *HBP1*, *MEF2A*, *PAFAH1B1*, *PPP1R3B*, *RPS6KA3* and *SESN1* were downregulated in BCa compared to NAT; meanwhile, *MAP3K2* and *YY1* were up-regulated. There were no statistical differences between BCa and NAT in *FRS2* and *STAT3*. We continued the analysis with the eight target genes that were down-regulated in BCa, as these could potentially be tumor suppressor genes.

### 2.5. Hsa-miR-106b-5p and miR-21-5p Are Upregulated in More Aggressive BCa Subtypes and Are Predictors of Worse Overall Survival

We evaluated the expression of hsa-miR-21-5p and miR-106b-5p in BCa from the TCGA BRCA dataset and compared them between subtypes and stages (Figure 5A). We found that hsa-miR-106b-5p was up-regulated in more aggressive subtypes (Basal), while hsa-miR-21-5p was up-regulated in intermediate aggressive subtypes (Luminal B and Her2). No significant differences were found in terms of the stage for these miRNAs.

Additionally, we performed Kaplan–Meier plots to evaluate the overall survival (OS) based on the expression of these miRNAs (Figure 5B) using a cohort of 1262 BCa patients. We found that patients who had a higher expression of hsa-miR-21 and miR-106b had lower survival and worse prognosis. Altogether these results propose hsa-miR-21-5p and miR-106b-5p as new biomarkers for BCa prognosis.

### 2.6. GAB1, GNG12, HBP1 and SESN1 Are Downregulated in More Aggressive BCa Subtypes and Could Be Used as Prognosis Biomarkers

We further analyzed the relevance of the eight selected miRNA-target genes from Figure 4B in terms of aggressiveness based on their expression in different BCa tissue subtypes (Figure 5C). We found that *GAB1*, *GNG12*, *HBP1* and *SESN1* expression was reduced in more aggressive compared with less aggressive subtypes, in concordance to the fact that hsa-miR-106b-5p was up-regulated in these tissues and could be repressing these genes. Moreover, we found that *MEF2A*, *PAFAH1B1*, *PPP1R3B* and *RPS6KA3* were downregulated in intermediate aggressive tumors (Luminal B and Her2) and up-regulated in Luminal A and Basal subtypes, suggesting that these genes could be regulated mostly by hsa-miR-21-5p.

We also performed Kaplan–Meier plots to determine OS and relapse-free survival (RFS) from the genes that were down-regulated in more aggressive subtypes in a cohort of 1879 BCa patients. As shown in Figure 6, patients with low tumor expression of *GNG12*, *HBP1* and *SESN1* genes presented reduced OS (Figure 6A) and worse RFS (Figure 6B), compared to patients with high expression of these genes. Moreover, low *GAB1* expression showed only worse RFS without changes in the OS. These results reinforce the relevant role for these genes as prognosis biomarkers in advanced BCa patients.

### 2.7. GAB1, GNG12, HBP1 and SESN1 Negatively Correlate with hsa-miR-106b-5p in BCa Tissues

To determine whether *GAB1*, *GNG12*, *HBP1* and *SESN1* could be regulated by hsa-miR-106b-5p in BCa tissue, we performed a correlation matrix using available datasets in TCGA BRCA. We found that *GAB1*, *GNG12*, *HBP1* and *SESN1* negatively correlated with hsa-miR-106b-5p, with rho values of −0.19, −0.38, −0.29 and −0.31, respectively (Figure 7A). Interestingly, *GAB1*, *GNG12* and *HBP1* genes positively correlated with *HBP1* with rho values of 0.4, 0.39 and 0.35, respectively (Figure 7A).

We plotted the expression of hsa-miR-106b-5p and each target gene to determine their correlation based on BCa subtype. As shown in Figure 7B, we found that more aggressive subtypes, such as Basal, had higher expression of hsa-miR-106b-5p and lower expression of the four target genes, in comparison with less aggressive subtypes, such as Luminal A, that had a lower expression of hsa-miR-106b-5p and higher expression of the target genes.

To determine the relevance of these genes in BCa metastasis, we used TNMplot in a cohort of 7893 subjects. We found that the expression of *GAB1*, *GNG12*, *HBP1* and *SESN1* was reduced in BCa compared to normal breast tissues (Figure 7C), as we previously showed in the TCGA cohort (Figure 4B). More importantly, these genes’ expression was dramatically diminished in metastatic tissues compared to normal breast tissues and BCa (Figure 7C).

## 3. Discussion

In this work, we provide new evidence of miRNAs and genes that could be used in the study of BCa aggressiveness and prognosis. We found that hsa-miR-21-5p and miR-106b-5p were both up-regulated in BCa tissue in mice and human samples. Accordingly, eight miRNA target-genes (*GAB1*, *GNG12*, *HBP1*, *MEF2A*, *PAFAH1B1*, *PPP1R3B*, *RPS6KA3* and *SESN1*) were downregulated in BCa compared to NAT tissue in patients. 

We found that hsa-miR-21-5p was up-regulated in BCa subtypes Her2+, correlated with worse OS and could be regulating *MEF2A*, *PAFAH1B1*, *PPP1R3B* and *RPS6KA3*, since they were mostly downregulated in Her2+ BCa subtypes. Moreover, we found that hsa-miR-106b-5p was up-regulated in more aggressive BCa subtypes and correlated with worse OS, and four of the eight target genes (*GAB1*, *GNG12*, *HBP1* and *SESN1*) were downregulated in more aggressive subtypes and correlated with worse OS and RFS. Finally, we found that their expression negatively correlated with hsa-miR-106b-5p expression in BCa tissue, proposing them as promising prognosis biomarkers.

There is compelling evidence proposing hsa-miR-21-5p as an oncomiR [17,18,19,20]. It is located in 17q23.2, which is a region that has been found to be amplified in breast carcinomas and contains several oncogenes [21]. This miRNA was found up-regulated in ductal carcinomas in situ (DCIS) compared to non-malignant breast tissues [21], in several cancers, including BCa [17,19,22,23] and in the circulation of BCa patients compared to healthy donors [21,22,23,24]. Moreover, it was found up-regulated in plasma from BCa patients with Her2 and Luminal B subtypes, which proposes a link between hsa-miR-21-5p and the Her2 receptor [25]. In particular, there is evidence of Her2 and hsa-miR-21-5p overexpression correlation, since hsa-miR-21-5p is up-regulated in Her2+ BCa [26], and it was also found associated to more aggressive BCa [20,27].

In terms of BCa progression, it was found to be related to BCa pathogenesis, proliferation, invasion, apoptosis, the epithelial to mesenchymal transition (EMT), cell cycle control and metastasis in tumor cells [18,19,20,22,23,25,26]. It was found an association between up-regulation of hsa-miR-21-5p and poor disease-free survival (DFS) in early stage BCa [20]; however, there is contradictory evidence for the correlation between hsa-miR-21-5p expression and OS [17,18]. Thus, these data, together with our findings, strongly support hsa-miR-21-5p as a biomarker for BCa diagnosis and prognosis.

Hsa-miR-106b-5p is a member of the 106b-25 cluster and a paralogous of the 17–92 cluster [28] that was found amplified and/or overexpressed in several tumors, including BCa [29,30,31]. It was up-regulated in the circulation of BCa patients compared to healthy donors [30] and in BCa compared to benign lesions [28,30]. In concordance to our findings, hsa-miR-106b-5p was found up-regulated in triple negative BCa compared to other subtypes [28]. The role of hsa-miR-106b-5p in BCa progression is highly reported: it was found associated to EMT [29,30,32,33,34], tumor progression [35], cell proliferation and migration [33,34], cell cycle control and cellular response to stress [30,36] and apoptosis and angiogenesis [29,31]. 

Preclinical studies demonstrated that hsa-miR-106b-5p can predict the presence of metastasis [30,36] since mice with up-regulation of this miRNA in tumors had developed lung metastasis [34] and increased number of metastatic nodules in the liver [35]. Moreover, hsa-miR-106b-5p was found up-regulated in metastatic primary tumors compared to non-metastatic, up-regulated in metastasis compared to primary tumors [35] and even up-regulated in secondary metastasis compared to primary metastasis [36]. 

In terms of BCa patient survival, it has been reported that those who have lower hsa-miR-106b-5p expression have longer DFS [35]. Moreover, in another study, BCa patients with higher expression of this miRNA had shorter DFS and OS as well as an augmented recurrence risk [30]. Higher levels of hsa-miR-106-5p were also associated to decreased tumor relapse [31]. In summary, all this evidence suggests hsa-miR-106-5p as a good biomarker to predict recurrence and progression and as a prognostic biomarker to identify local or distant recurrence and high-risk BCa.

In this work, we found that hsa-miR-106b-5p and miR-21-5p share several target genes, including *GAB1*, *GNG12*, *HBP1* and *SESN1*, that were down-regulated in breast primary tumors and metastases. According to Lee et al. [37], *GNG12* was found down-regulated in local recurrent DCIS compared to no recurrent, which strongly supports our data. 

Furthermore, *SESN1* was found to be mainly involved in metabolism regulation and aging via the AMPK/mTOR pathway [38]. In particular, it interacts with AMPK in the stress response, tumorigenesis suppression and maintenance of genomic integrity [38]. It was found that mutant p53 blockades SESN1/AMPK complex, unleashing oncogenic effects such as inhibition of apoptosis, increasing drug resistance and proliferation [39]. It was found that, in patients with mutant p53, a lower expression of several genes, including *SESN1*, correlated with lower RFS times and distant-metastasis-free survival [40].

*GAB1*’s role in BCa has been largely studied and, in opposition with our findings, it has been proposed as an oncogene. GAB1 is a key scaffold protein that enhances the downstream signaling of c-Met among others tyrosine-kinase receptors [41,42], it is involved in the formation of invadopodia [41], allows epidermal growth factor receptor (EGFR) dimers to directly signal PI3K [43] and is implicated in EGF-induced activation of MAPK [44].

In opposition with our findings, *GAB1* was found up-regulated in breast tumors compared to benign mammary hyperplasia from patients [45]. Moreover, *GAB1* overexpression was associated to metastasis in Her2 and triple negative BCa subtypes [45]. However, this study was limited to a low number of subjects, which could be the cause of disagreement with our results. 

Several studies have proposed *HBP1* as a transcriptional repressor, since it has an HMG box DNA-binding domain that allows it to impair the union between DNA and active transcription factors [46,47,48,49]. *HBP1* is located in chromosome 7q31.1, a region that is found usually deleted in several cancer types [46,50] and 30% of BCa patients have mutants or variants of *HBP1* [47,48]. In BCa preclinical models, *HBP1* depletion increased cell cycle progression, proliferation, invasion, migration, and tumorigenesis [47,49]. 

Additionally, *HBP1* was found down-regulated in breast tumors compared to normal tissue from patients, which was associated with poor prognosis, RFS and relapse [46,47,50]. Furthermore, hsa-miR-17-5p, a member of the 17-92 cluster, paralogous of the 106b-25 cluster, binds to *HBP1*, lowering its expression [50]. These findings could indicate that hsa-miR-106b-5p could also be involved in *HBP1* decrease; however, further studies are needed to confirm our hypothesis.

In summary, all these data suggest that *GAB1*, *GNG12*, *HBP1* and *SESN1* act as tumor suppressor genes by the proposed mechanism indicated in Figure 8. As shown, less aggressive BCa subtypes and NAT display low expression of hsa-miR-106b-5p, which, in turn, maintains *GAB1*, *GNG12*, *HBP1* and *SESN1* high expression. However, more aggressive BCa correlates with hsa-miR-106b-5p up-regulation and the down-regulation of its targets *GAB1*, *GNG12*, *HPB1* and *SESN1*, unleashing BCa aggressiveness and, therefore, worsening the prognosis of BCa patients.

## 4. Materials and Methods

### 4.1. Cell Culture

The 4T1 murine cell line (ATCC: CRL-2539) was cultured in RPMI 1640 (Invitrogen) supplemented with 10% of fetal bovine serum and antibiotics in a 5% CO_2_ humidified atmosphere at 37 °C.

### 4.2. BCa Allograft Murine Model

We housed 16-week-old Balb/c female mice (*n* = 16) under pathogen-free conditions following the IBYME’s animal care guidelines. Mice were randomized into two groups: Control and BCa. BCa mice were inoculated in the mammary fat pad with 1 × 10^4^ 4T1 cells. Control animals were not inoculated at any point of the experiment. Control and BCa were sacrificed both at the same time (4 weeks after the inoculation) and tumor and mammary gland (MG) samples were collected. Tumor measurement was done as previously described [51]. 

### 4.3. RNA Isolation and RT-qPCR Analysis

The total RNA from allografts and MG were isolated using TriReagent (Molecular Research Center). Stem-loop RT-qPCR method was used to retrotranscribe miRNAs as previously described [52,53,54]. Briefly, 100 ng of total RNA and 0.07 μM of stem-loop specific primer were preheated at 70 °C for 5 min. Then, retrotranscription was performed using M-MLV RT (Promega) and incubated in TC 9639 Thermal Cycler (Benchmark) at 16 °C for 30 min, 42 °C for 60 min and 70 °C for 2 min. qPCRs were performed in StepOne Plus Real Time PCR (Applied Biosystems) using Taq DNA Polymerase (Pegasus) as previously described [51]. 

All reactions were run in duplicate. The expression levels of miRNAs were calculated using the ΔΔCt method normalizing to mmu-miR-191-5p and the Control. The primer sequences for Stem-loop RT-qPCR are listed in Table 1. The results are given as the median and interquartile ranges. Data normalization and homogeneity of variances was assessed using the Shapiro–Wilk test, F test or boxplot, respectively. Student’s *t*-test was applied for data that fulfill the requirements. Otherwise, the Welch, Wilcoxon or Median test was performed. We used a significance level of 5%.

### 4.4. TCGA Dataset Analysis

Clinical-pathological data, mature miRNA and gene expression of breast cancer (BCa) (*n* = 1097) and normal adjacent tissue (NAT) (*n* = 114) of patients, and normal mammary gland (NMG) (*n* = 80) was obtained from de TCGA Breast Cancer (BRCA) cohort or from the GTEx project available in the UCSC Xena bioinformatics tool (https://xena.ucsc.edu/, accessed on 15 August 2021) [16]. The miRNA-Seq (IlluminaHiSeq_miRNASeq) and RNA-Seq (IlluminaHiSeq) data was downloaded as log_2_ (RPM + 1) values. Seventy-five BCa samples paired with 75 NAT were included in the present study. 

When miRNAs were analyzed, 491 BCa samples were used, and, when genes were analyzed, 821 BCa samples were used. Data normalization and homogeneity of variances was assessed using the Shapiro–Wilk test, and Levene, F test or boxplot, respectively. A paired sample *t*-test was applied for data that fulfill the requirements. Otherwise, the Wilcoxon test was performed. When three or more groups were analyzed, one-way ANOVA followed by Tukey was performed for data that fulfill the requirements. Otherwise, the Kruskal–Wallis followed by Dunn’s Test was performed. We used a significance level of 5%.

### 4.5. Functional Enrichment Analysis

To identify experimentally validated target (EVT) genes regulated by differentially expressed miRNAs in human BCa and mice allografts selected miRNAs, we employed the DIANA TARBASE v8 resource (threshold score > 0.5) [55]. Functional enrichment analyses of the obtained gene lists were performed using the ClueGo Cytoscpape’s plug in and the Enrichr resource. For pathway terms and annotation, we used those provided by KEGG and BioPlanet (http://tripod.nih.gov/bioplanet/, accessed on 1 August 2021; https://www.genome.jp/kegg/pathway.html, accessed on 1 August 2021). For network construction of the interactions miRNA-target-pathways, we used a Sankey diagram.

### 4.6. Principal Component Analysis (PCA)

Publicly available gene expression values for TCGA Breast Cancer (BRCA) and the GTEx project patient samples were downloaded as previously described. PCA was performed to determine samples distribution based on the expression of 51 common target genes from up-regulated miRNAs in both breast human tissue and mice allografts. For PCA plots, the R function “prcomp” from stats package (version 4.0.2) was used.

### 4.7. Single-Sample Gene-Set Enrichment Analysis

A ssGSEA was performed to analyze the coordinated regulation of a defined gene signature and the activation of specific biological processes in BCa samples. Log_2_ (RPM + 1) gene expression values of 474 BCa samples were obtained from TCGA-BRCA cohort and loaded into GenePattern web-tool (https://www.genepattern.org, accessed on 30 August 2021). A gene set enrichment profile was obtained for each sample using a 51 gene signature obtained from common target genes from up-regulated miRNAs in both breast human tissue and mice allografts. Then, Spearman correlation analysis was performed between the expression of two up-regulated miRNAs and the enrichment score obtained from the ssGSEA (GraphPad Prism 8.0.1; Prism, San Diego, CA, USA).

### 4.8. Correlation Matrix

Expression levels of miRNA and target genes in breast tumors from patients and their PAM50 classification were obtained from TCGA BRCA data available in UCSC Xena. Only BCa samples were included in the present analysis. Using the Hmisc R package, we generated a correlation matrix for each miRNA, applying the Spearman correlation coefficient.

### 4.9. Kaplan–Meier Plots

Overall Survival (OS) was performed for the selected miRNAs using data from miRpower bioinformatic tool (https://kmplot.com, accessed on 3 September 2021) [56]. OS and Relapse Free Survival (RFS) for the selected genes were performed using data from the Kaplan–Meier plotter bioinformatic tool (https://kmplot.com, accessed on 3 September 2021) [57]. Data was analyzed and plotted using the libraries survminer and survival in R software. 

### 4.10. Normal, Tumor and Metastatic Gene Expression

Data and plots comparing normal, tumor and metastatic gene expression were performed using TNMplot bioinformatic tool (https://TNMplot.com, accessed on 1 September 2021) [58], which contains data from 242 normal tissues, 7569 BCa tissues and 82 metastatic tissues. For comparisons among tissues, Kruskal–Wallis followed by Dunn’s Test was performed.

## Figures and Tables

**Figure 1 ijms-22-11135-f001:**
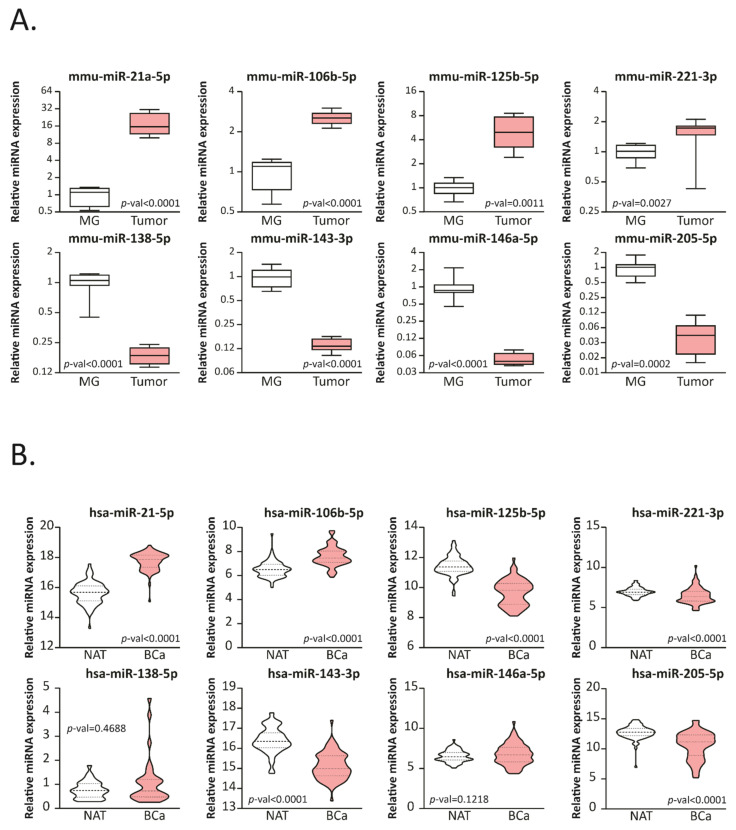
miRNAs expression is modulated in BCa mice and human samples. (**A**) Stem-loop RT-qPCR from 4T1 allografts (Tumor) or mammary-glands (MG) Balb/c mice using specific primers for the indicated miRNAs is shown (*n* = 8 per group; primers were run in duplicates). The data were normalized to mmu-miR-191-5p and MG. Statistical analysis was performed using a *T*-test, Welch or sing-median test when appropriate. (**B**) Expression levels of the corresponding miRNAs in BCa tumors paired normal adjacent tissue (NAT) from the TCGA-BRCA dataset are graphed in read per millions. The name of the miRNAs is indicated up each plot in bold. Statistical analysis was performed using a paired *T*-test or Wilcoxon test when appropriate.

**Figure 2 ijms-22-11135-f002:**
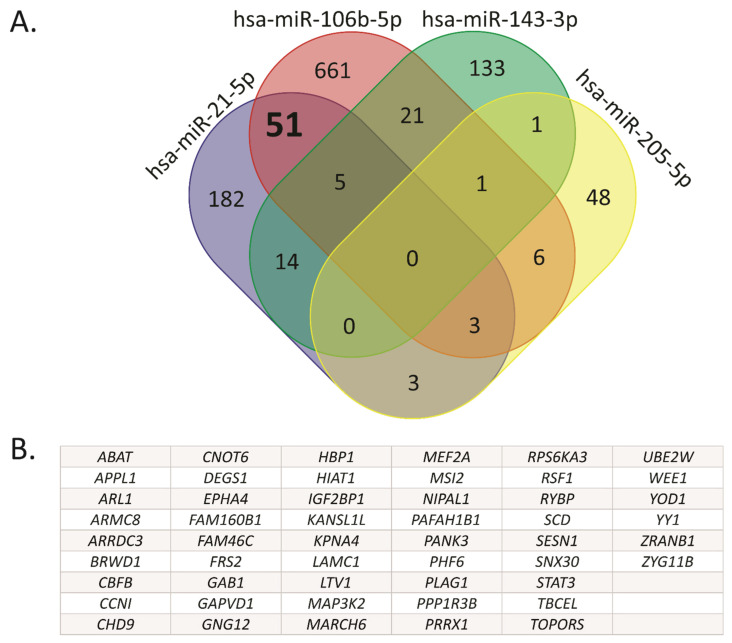
Common target genes between hsa-miR-106b-5p and miR-21-5p. (**A**) The Venn diagram was performed using validated target genes from hsa-miR-106b-5p, miR-21-5p, miR-143-3p and miR-205-5p. (**B**) The 51 target genes shared exclusively between hsa-miR-106b-5p and miR-21-5p, but not with hsa-miR-143-3p and miR-205-5p, are listed.

**Figure 3 ijms-22-11135-f003:**
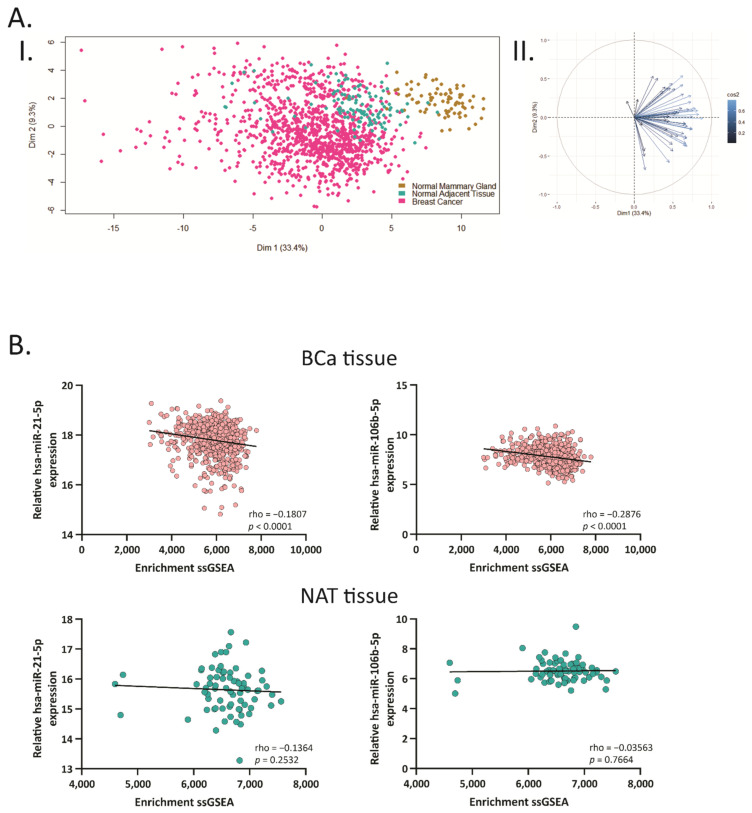
Principal component analysis showing correlation between hsa-miR-106b-5p and miR-21-5p with their target genes. (**A**) I. Scatterplot of the two principal components of principal component analysis (PCA) from the target genes expression data. The golden, blue and pink circles represent normal mammary gland (NMG), normal adjacent tissue (NAT) and BCa samples, respectively. II Biplot representing the common genes and their relevance in each dimension of the plot. (**B**) Spearman correlation between the ssGSEA enrichment scores calculated based on the 51 target genes and mature miRNA expression in BCa tissue and NAT from the TCGA-BRCA dataset.

**Figure 4 ijms-22-11135-f004:**
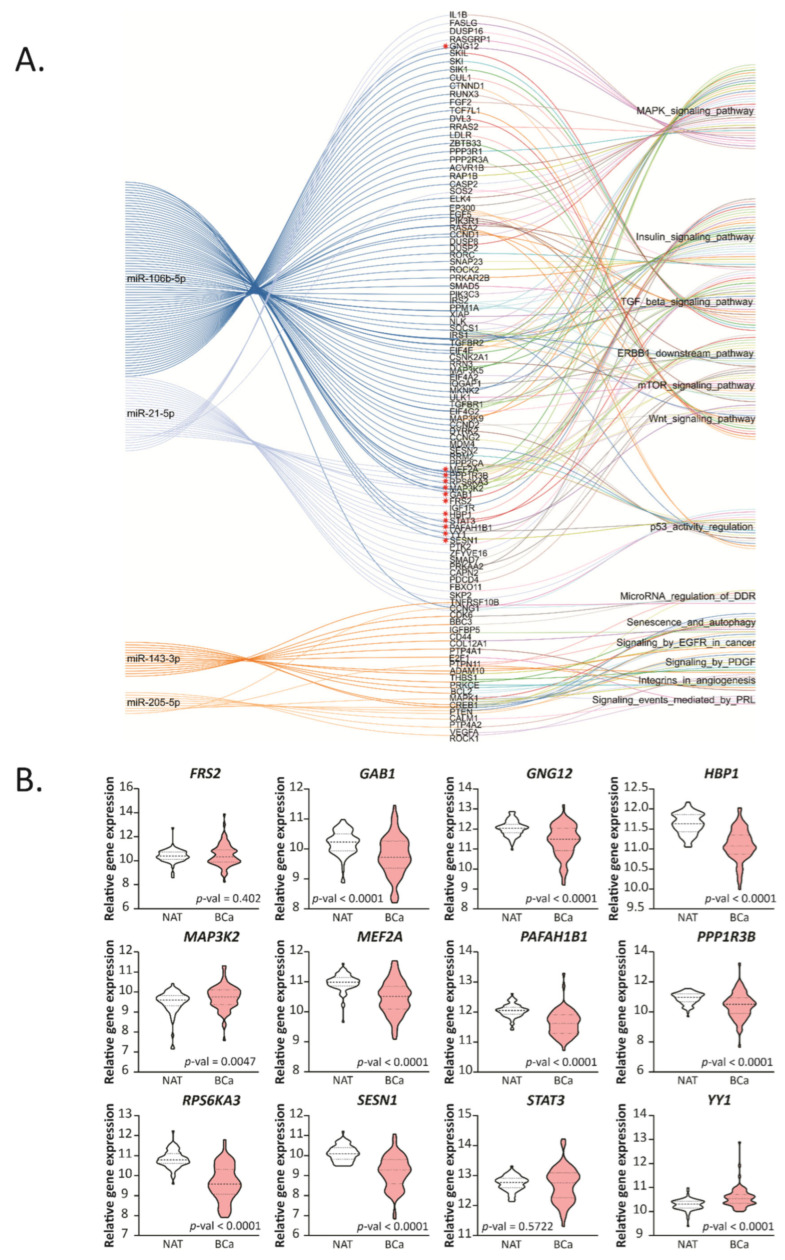
Functional enrichment of validated miRNA targets and their expression in human samples. (**A**) String diagram performed based on the top five significant KEGG pathways associated to the target genes of the up- and down-regulated miRNAs in common between humans and mice. Relevant genes related to the up-regulated miRNAs are marked with a red asterisk. (**B**) Expression levels of the selected genes in BCa tissue paired with normal adjacent tissue (NAT) from the TCGA-BRCA dataset are graphed in read per millions. The name of the genes is indicated up each plot in bold. Statistical analysis was performed using paired *T*-test or Wilcoxon test when corresponded.

**Figure 5 ijms-22-11135-f005:**
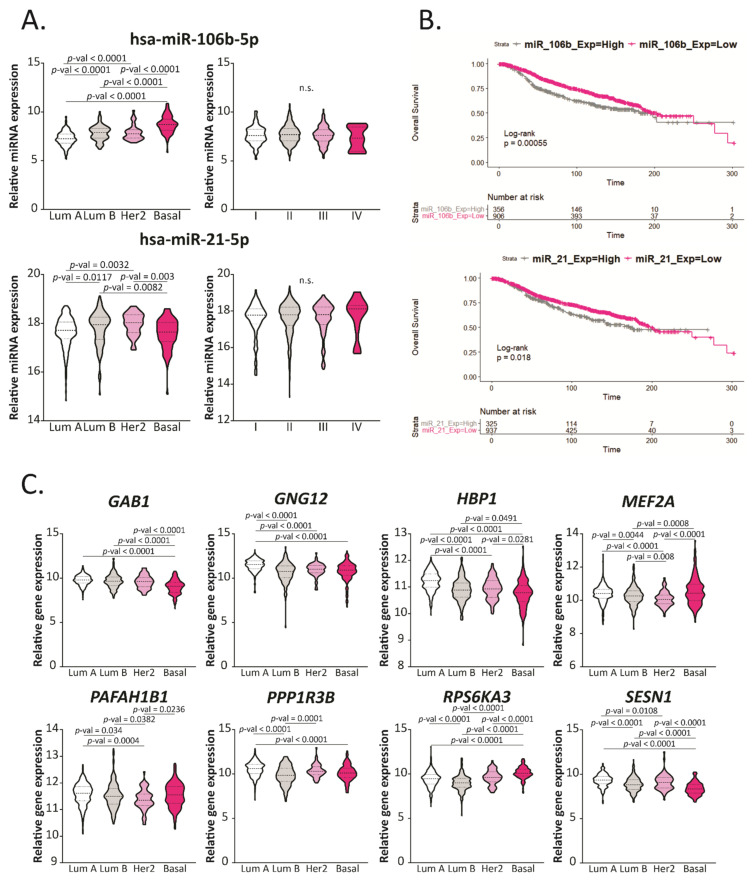
Hsa-miR-106b-5p, miR-21-5p and their target genes are deregulated in aggressive BCa subtypes and correlate with overall survival. (**A**) Expression levels of hsa-miR-106b-5p and miR-21-5p in BCa tissue divided in subtypes (Luminal A, Luminal B, Her2 or Basal) or Stages (I to IV) from the TCGA-BRCA dataset are graphed in read per millions. The name of the miRNAs is indicated up each plot in bold. Statistical analysis was performed using One-Way ANOVA or Kruskal–Wallis test followed by Tukey or Dunn’s, respectively, when corresponded. (**B**) Kaplan–Meier curves comparing the overall survival (OS) between BCa patients with high or low expression of hsa-miR-106b-5p or miR-21-5p, respectively, in tumors. Statistical analysis was performed using the Cox–Mantel test. (**C**) Expression levels of the mentioned genes in BCa tissue divided in subtypes (Luminal A, Luminal B, Her2 or Basal) from the TCGA-BRCA dataset are graphed in read per millions. The name of the miRNAs is indicated up each plot in bold. Statistical analysis was performed using One-Way ANOVA or the Kruskal–Wallis test followed by Tukey or Dunn’s, respectively, when corresponded.

**Figure 6 ijms-22-11135-f006:**
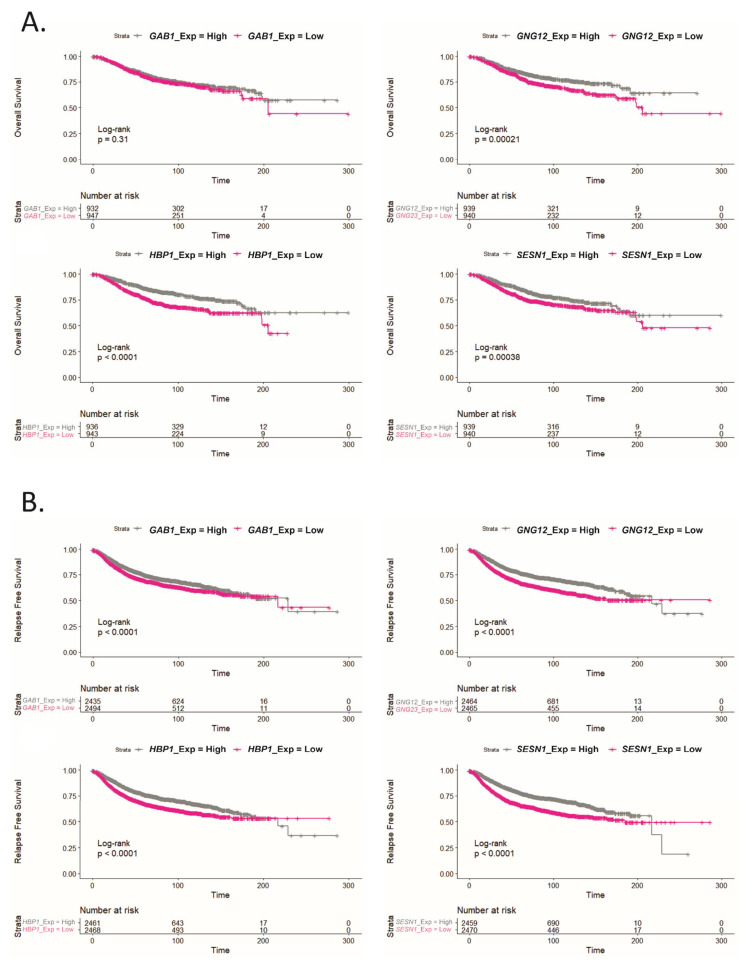
Lower expression of *GAB1*, *GNG12*, *HBP1* and *SESN1* correlates with worse relapse free survival and with overall survival. (**A**) Kaplan–Meier curve comparing the overall survival (OS) between BCa patients with high or low expression of the mentioned genes in tumors. Statistical analysis was performed using the Cox–Mantel test. (**B**) Kaplan–Meier curve comparing relapse free survival (RFS) between BCa patients with high or low expression of the mentioned genes in tumors. Statistical analysis was performed using the Cox–Mantel test.

**Figure 7 ijms-22-11135-f007:**
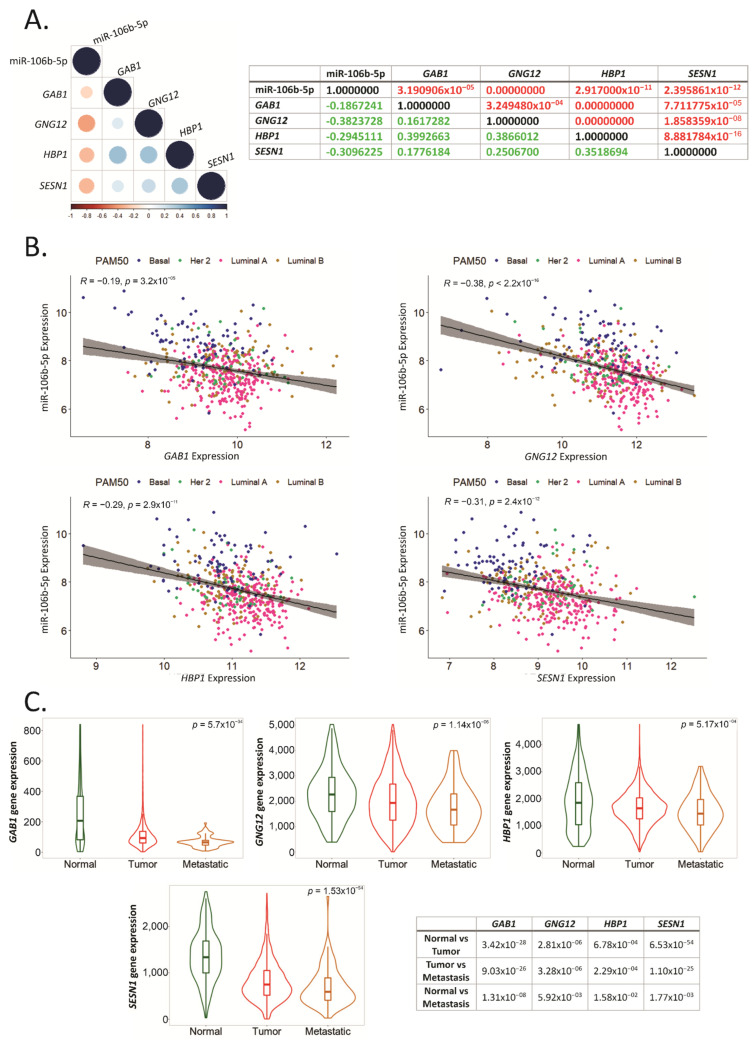
Hsa-miR-106b-5p negatively correlates with *GAB1*, *GNG12*, *HBP1* and *SESN1*. (**A**) Matrix correlation showing existing correlation between hsa-miR-106b-5p expression in BCa tissue from the TCGA-BRCA cohort and target genes. *p*-values and Spearman correlation value are indicated in the table in red and green, respectively. (**B**) Spearman correlation between hsa-miR-106b-5p expression in BCa tissue from the TCGA-BRCA dataset, and each target gene is plotted distinguishing between BCa subtypes. The pink, golden, green and blue circles represent Luminal A, Luminal B, Her2 and Basal BCa subtypes, respectively. (**C**) Expression levels of the mentioned genes in normal mammary glands (Normal), BCa tissue (Tumor) and metastatic BCa (Metastatic) are graphed using TNMplot bioinformatic tool. Statistical analysis was performed using the Kruskal–Wallis (K-W) test followed by Dunn’s. The *p*-value of the K-W test is indicated on the top right corner of each graph. Dunn’s comparison’s *p*-value corresponding to each gene and each graph are resumed in the table.

**Figure 8 ijms-22-11135-f008:**
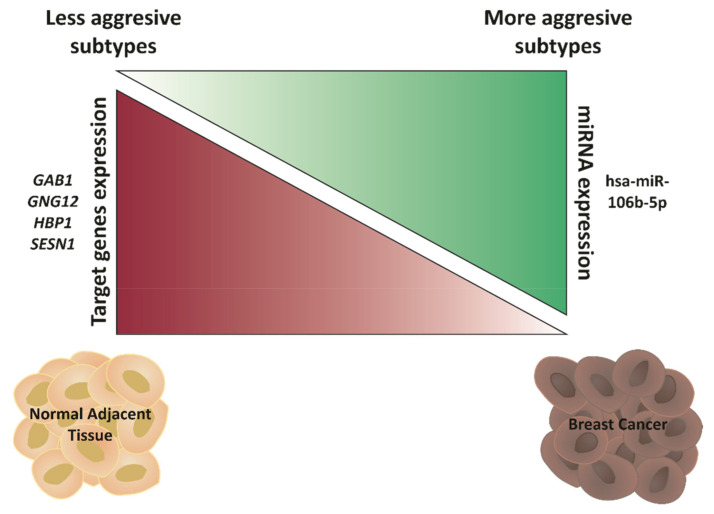
Hypothetical model. Hsa-miR-106b-5p is down-regulated in less aggressive BCa subtypes and NAT, which keeps *GAB1*, *GNG12*, *HBP1* and *SESN1* up-regulated. When BCa progresses, or in more aggressive BCa subtypes, hsa-miR-106b-5p is then up-regulated and targets *GAB1*, *GNG12*, *HPB1* and *SESN1*, diminishing their expression, unleashing BCa aggressiveness and, therefore, resulting in a worse BCa prognosis.

**Table 1 ijms-22-11135-t001:** Primer sequences used for stem-loop RT-qPCR.

Primer	Sequence (5′–3′)	T_ann_ (°C)
RT-Stem-loop-Rv	TGGTGCAGGGTCCGAGGTATT	–
RT-mmu-miR-21a-5p-STEM	GTCTCCTCTGGTGCAGGGTCCGAGGTATTCGCACCAGAGGAGACTCAACA	–
RT-mmu-miR-21a-5p Fw	CGGGGGGTAGCTTATCAGACTG	65
RT-mmu-miR-106b-5p-STEM	GTCTCCTCTGGTGCAGGGTCCGAGGTATTCGCACCAGAGGAGACATCTGC	–
RT-mmu-miR-106b-5p Fw	GCGGCGGTAAAGTGCTGACAG	67
RT-mmu-miR-125b-5p-STEM	GTCTCCTCTGGTGCAGGGTCCGAGGTATTCGCACCAGAGGAGACTCACAA	–
RT-mmu-miR-125b-5p Fw	CCGCCTCCCTGAGACCCTAAC	65
RT-mmu-miR-221-3p-STEM	GTCTCCTCTGGTGCAGGGTCCGAGGTATTCGCACCAGAGGAGACGAAACC	–
RT-mmu-miR-221-3p Fw	GGCGGAGCTACATTGTCTGCTG	65
RT-mmu-miR-138-5p-STEM	GTCTCCTCTGGTGCAGGGTCCGAGGTATTCGCACCAGAGGAGACCGGCCT	–
RT-mmu-miR-138-5p Fw	GGCGGAGCTGGTGTTGTGAATC	67
RT-mmu-miR-143-3p-STEM	GTCTCCTCTGGTGCAGGGTCCGAGGTATTCGCACCAGAGGAGACGAGCTA	–
RT-mmu-miR-143-3p Fw	GGGCGGTGAGATGAAGCACTG	67
RT-mmu-miR-146a-5p-STEM	GTCTCCTCTGGTGCAGGGTCCGAGGTATTCGCACCAGAGGAGACAACCCA	–
RT-mmu-miR-146a-5p Fw	CGGGCGGTGAGAACTGAATTCC	65
RT-mmu-miR-205-5p-STEM	GTCTCCTCTGGTGCAGGGTCCGAGGTATTCGCACCAGAGGAGACCAGACT	–
RT-mmu-miR-205-5p Fw	CGCGTCCTTCATTCCACCGG	65
RT-mmu-miR-191-5p-STEM	GTCTCCTCTGGTGCAGGGTCCGAGGTATTCGCACCAGAGGAGACCAGCTG	–
RT-mmu-miR-191-5p Fw	GCGGCAACGGAATCCCAAAAG	70

## Data Availability

Publicly available datasets were analyzed in this study. This data can be found here: https://xena.ucsc.edu/, accessed on 15 August 2021; http://tripod.nih.gov/bioplanet/, accessed on 1 August 2021; https://www.genome.jp/kegg/pathway.html, accessed on 1 August 2021; https://kmplot.com, accessed on 3 September 2021; https://TNMplot.com, accessed on 1 September 2021.

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
