# Peer review of "MiR-106b-5p: A Master Regulator of Potential Biomarkers for Breast Cancer Aggressiveness and Prognosis"

_ijms, 2021, doi:10.3390/ijms222011135_

Round 1
Reviewer 1 Report
The authors have comprehensive and logical study investigating miRNAs as potential biomarkers for breast cancer prognosis. They have described a thorough bioinformatic approach to assess the effect of miRNA mis-regulation in breast cancer to expression of predicted target genes, the pathways affected and correlated this with tumours stage, grade and patient survival.
I only have minor comments:
Figure 1: Every graph included in this figure has a scale on the Y -axis making this confusing for the reader, especially as some of these values have up to 7 digits after the decimal point. To clean these graphs up I would also normalise each to the MG negative control and therefore the fold difference to the tumour tissue is comparable between the different miRNAs
Figure 1B, 3B, 4B, 5A, 5C – the labels for the Y axis are not sufficient for reader understanding.
Figure 7C: what does the P value indicated in the top right hand corner of each graph indicate? It is not explained, nor does it correspond to any figure in the table – I assume the table is showing P values of the samples although this is not explained either.
MG (initially used in line 74) is not defined.
EVT (initially used in line 98) is not defined
MNG (initially used in line 113) is not defined.
ssGSEA (initially used in line 119) is not defined.
OS (initially used in line 167) is not defined.
RFS (initially used in line 193) is not defined.
There are some typos and english grammar errors sporadically throughout the manuscript.
The references are very inconsistent - e.g. authors names are not included in many references
Author Response
Thank you for the clear review of our manuscript entitled “MiR-106b-5p: a master regulator of potential biomarkers for breast cancer aggressiveness and prognosis.”
We have reviewed all of the referee’s comments and have made additions and changes to the manuscript accordingly. As a result, we feel that the manuscript has been improved. We thank the reviewers for their thoughtful review.
A point by point response to the comments is given below:
1- Figure 1: Every graph included in this figure has a scale on the Y -axis making this confusing for the reader, especially as some of these values have up to 7 digits after the decimal point. To clean these graphs up I would also normalise each to the MG negative control and therefore the fold difference to the tumour tissue is comparable between the different miRNAs
Response: Thank you for your suggestion. We have normalized all the graphs to MG negative control. You will be able to see the changes in the new version of the manuscript attached below.
2- Figure 1B, 3B, 4B, 5A, 5C – the labels for the Y axis are not sufficient for reader understanding.
Response: Thank you for your suggestion. We have changed all the Y axis and completed the legends in order to be easier to understand
3- Figure 7C: what does the P value indicated in the top right hand corner of each graph indicate? It is not explained, nor does it correspond to any figure in the table – I assume the table is showing P values of the samples although this is not explained either.
Response: Thanks for your observation. The P-value indicated in the top right corner of each graph is the general value of the Kruskal-Wallis test. We have added this information to the figure legend. We have added to the figure legend the explanation of the table showing p-values as well
4- MG (initially used in line 74) is not defined.
Response: Thanks for the observation. We have changed it in the new version of the manuscript.
5- EVT (initially used in line 98) is not defined
Response: Thanks for the observation. We have changed it in the new version of the manuscript.
6- MNG (initially used in line 113) is not defined.
Response: Thanks for the observation. We have changed it in the new version of the manuscript.
7- ssGSEA (initially used in line 119) is not defined.
Response: Thanks for the observation. We have changed it in the new version of the manuscript.
8- OS (initially used in line 167) is not defined.
Response: Thanks for the observation. We have changed it in the new version of the manuscript.
9- RFS (initially used in line 193) is not defined.
Response: Thanks for the observation. We have changed it in the new version of the manuscript.
10- There are some typos and english grammar errors sporadically throughout the manuscript.
Response: Thanks for the observation. We have edited the entire manuscript; you can see the changes in the attached new version.
11- The references are very inconsistent - e.g. authors names are not included in many references
Response: We are sorry for not noticing the format mistake; we have changed it in the new version of the manuscript.
Reviewer 2 Report
Nice Manuscript. However, i have the following comments:
Major comments:
1- the Authors stated in many occasions that "51 target genes were overlapped between hsa-miR-21-5p and miR-106b-5p, 105 listed in Figure 2B ". In fact the target genes shared between hsa-miR-106b-5p and miR-21-5p are 59 (51+5+0+3) as shown in figure 2-A not 51 as the authors mentioned in the entire manuscript (e.g., Line 22, 104-105, Figure 2-B, Line 112, 115, 121, 132, 423, 431,....etc). Please revise the analysis based on these 59 genes. Also, please revise figure 2-b and the entire manuscript in accordance.
2- Again the Authors stated that "There was only one target gene in common between hsa-miR-143-3p and miR-205- 104 5p". In fact there is two (1+1+0+0) target genes in common as shown in figure 2-A
Author Response
Thank you for the clear review of our manuscript entitled “MiR-106b-5p: a master regulator of potential biomarkers for breast cancer aggressiveness and prognosis.”
We have reviewed all of the referee’s comments and have made additions and changes to the manuscript accordingly. As a result, we feel that the manuscript has been improved. We thank the reviewers for their thoughtful review.
A point by point response to the comments is given below:
- the Authors stated in many occasions that "51 target genes were overlapped between hsa-miR-21-5p and miR-106b-5p, 105 listed in Figure 2B ". In fact the target genes shared between hsa-miR-106b-5p and miR-21-5p are 59 (51+5+0+3) as shown in figure 2-A not 51 as the authors mentioned in the entire manuscript (e.g., Line 22, 104-105, Figure 2-B, Line 112, 115, 121, 132, 423, 431,....etc). Please revise the analysis based on these 59 genes. Also, please revise figure 2-b and the entire manuscript in accordance.
Response: Thanks for your important suggestion. Our first analysis included all the 59 genes (data not shown); however, in this work we focused on the hsa-miR-106b-5p and miR-21-5p target genes excluding those that could be related to hsa-miR-143-3p and miR-205-5p. Thus, our analysis is addressed only to the miR-106b-5p and miR-21-5p target genes.
We have added to our manuscript why we choose only the 51 genes and not all the 59, as follows: “We obtained a list of experimentally validated target (EVT) genes for hsa-miR-21-5p, miR-106b-5p, miR-205-5p and miR-143-3p, using DIANA TARBASE v8 resource. To identify common target genes, we used Venn diagrams (Figure 2A, Table S1). Based on this analysis, two miRNAs-target gene sets were obtained. The first one included the target genes modulated exclusively by the up-regulated miRNAs hsa-miR-21-5p and miR-106b-5p, excluding those 8 genes shared with hsa-miR-143-3p and miR-205-5p.”
- Again the Authors stated that "There was only one target gene in common between hsa-miR-143-3p and miR-205- 104 5p". In fact there is two (1+1+0+0) target genes in common as shown in figure 2-A
Response: Thanks for your observation. Again, we focused on the validated target genes from hsa-miR-143-3p and miR-205-5p excluding the ones that were targets from neither hsa-miR-106b-5p nor miR-21-5p.
We added this information to the new version of the manuscript, as follows: “The second one included the genes modulated exclusively by the down-regulated miRNAs hsa-miR-143-3p and miR-205-5p, excluding the gene shared with hsa-106b-5p. There was only one target gene in common between hsa-miR-143-3p and miR-205-5p, while 51 target genes were overlapped between hsa-miR-21-5p and miR-106b-5p, listed in Figure 2B. We based the rest of the analysis on this 51-gene signature.”
You may see all the changes in the attached version of the manuscript above
Round 2
Reviewer 2 Report
The Authors addressed all my comments